# Hsp70—A Universal Biomarker for Predicting Therapeutic Failure in Human Female Cancers and a Target for CTC Isolation in Advanced Cancers

**DOI:** 10.3390/biomedicines11082276

**Published:** 2023-08-16

**Authors:** Alexia Xanthopoulos, Ann-Kathrin Samt, Christiane Guder, Nicholas Taylor, Erika Roberts, Hannah Herf, Verena Messner, Anskar Trill, Katharina Larissa Kreszentia Holzmann, Marion Kiechle, Vanadin Seifert-Klauss, Sebastian Zschaeck, Imke Schatka, Robert Tauber, Robert Schmidt, Katrin Enste, Alan Graham Pockley, Dominik Lobinger, Gabriele Multhoff

**Affiliations:** 1Center for Translational Cancer Research TU München (TranslaTUM), Klinikum rechts der Isar, Technical University of Munich (TUM), 81675 Munich, Germany; alexia.xanthopoulos@gmx.at (A.X.); ann-kathrin@familie-samt.de (A.-K.S.); christiane.guder@tum.de (C.G.); nicholasjosef.taylor@gmail.com (N.T.); erika.roberts@tum.de (E.R.); hannah.herf@tum.de (H.H.); verena.messner@tum.de (V.M.); anskar.trill@googlemail.com (A.T.);; 2Department of Gynecology and Obstetrics, Klinikum rechts der Isar, Technical University of Munich (TUM), 81675 Munich, Germany; marion.kiechle@tum.de (M.K.); vanadin.seifert-klauss@tum.de (V.S.-K.); 3Department of Radiation Oncology and Radiotherapy, Charité Berlin, 10117 Berlin, Germany; sebastian.zschaeck@charite.de; 4Department of Nuclear Medicine, Charité Berlin, 10117 Berlin, Germany; imke.schatka@charite.de; 5Department of Urology, Klinkum rechts der Isar, Technical University of Munich (TUM), 81675 Munich, Germany; robert.tauber@tum.de; 6Krankenhaus für Naturheilweisen, 81545 Munich, Germany; schmidt.robert@kfn-muc.de (R.S.); enste@kfn-muc.de (K.E.); 7John van Geest Cancer Research Centre, School of Science and Technology, Nottingham Trent University, Nottingham NG11 8NS, UK; graham.pockley@ntu.ac.uk; 8Department of Thoracic Surgery, München Klinik Bogenhausen, Lehrkrankenhaus der TU München, 81925 Munich, Germany; dominiklobinger@web.de; 9Department of Radiation Oncology, Klinikum rechts der Isar, Technical University of Munich (TUM), 81675 Munich, Germany

**Keywords:** Hsp70, liquid biopsy, tumor biomarker, circulating tumor cells, breast cancer, endometrial cancer, prostate cancer, head and neck cancer, lung cancer

## Abstract

Heat shock protein 70 (Hsp70) is frequently overexpressed in many different tumor types. However, Hsp70 has also been shown to be selectively presented on the plasma membrane of tumor cells, but not normal cells, and this membrane form of Hsp70 (mHsp70) could be considered a universal tumor biomarker. Since viable, mHsp70-positive tumor cells actively release Hsp70 in lipid micro-vesicles, we investigated the utility of Hsp70 in circulation as a universal tumor biomarker and its potential as an early predictive marker of therapeutic failure. We have also evaluated mHsp70 as a target for the isolation and enumeration of circulating tumor cells (CTCs) in patients with different tumor entities. Circulating vesicular Hsp70 levels were measured in the peripheral blood of tumor patients with the compHsp70 ELISA. CTCs were isolated using cmHsp70.1 and EpCAM monoclonal antibody (mAb)-based bead approaches and characterized by immunohistochemistry using cytokeratin and CD45-specific antibodies. In two out of 35 patients exhibiting therapeutic failure two years after initial diagnosis of non-metastatic breast cancer, progressively increasing levels of circulating Hsp70 had already been observed during therapy, whereas levels in patients without subsequent recurrence remained unaltered. With regards to CTC isolation from patients with different tumors, an Hsp70 mAb-based selection system appears superior to an EpCAM mAb-based approach. Extracellular and mHsp70 can therefore serve as a predictive biomarker for therapeutic failure in early-stage tumors and as a target for the isolation of CTCs in various tumor diseases.

## 1. Introduction

Comprising approximately 30% of all cancer cases, breast cancer remains the most common tumor in women, with an estimated worldwide incidence of 2.3 million cases in 2020 [1]. The local therapy of patients with non-metastatic breast cancer includes surgical removal of the tumor and potentially regional lymph nodes, in combination with adjuvant radiotherapy to prevent recurrence [2]. Depending on the age of the patient, tumor size, grading, lymph node, hormone receptor status, menopausal status, and HER2 expression, standard neoadjuvant or adjuvant treatment can involve anti-hormone, chemo- and/or antibody-based therapies [3,4]. Despite significant progress in the development of these treatment options, the monitoring of therapeutic response in breast cancer still requires improvement. Biomarkers that can be used to better monitor the effectiveness of an individual therapy and also predict disease recurrence earlier will undoubtedly improve patient management, overall outcomes and help to reduce normal tissue toxicities.

Prostate cancer is the second most common cancer in men worldwide and the most frequent cancer in Western countries. Despite advances in multimodal local and systemic treatment options, there is currently no cure for metastatic prostate cancer [5]. Transcriptomic analysis of plasma-derived exosomes and CTC counts appear to provide biomarkers that predict chemotherapy resistance and overall survival [5,6]. As Heat Shock Proteins (HSPs) of the HSP70 and HSP90 families play crucial roles in tumorigenesis, cytoprotection, epithelial-to-mesenchymal transition (EMT), invasion, and metastasis, inhibitors of these HSP families have been designed to render tumor cells more responsive to therapeutic interventions [7].

Herein, we assessed the utility of the major stress-inducible Hsp70 as a universal biomarker for predicting tumor responses in early-stage tumors and as a target for more efficient isolation of CTCs in advanced tumors [8]. Among all stress proteins, the 70 kDa family is the most highly conserved and best-studied group, consisting of 13 isotypes that can be distinguished by their amino acid sequence, expression levels, functions, and subcellular localization [8]. Members of the HSP70 family reside in nearly all cellular compartments, such as the nucleus, cytosol, mitochondria, and lysosomes, as well as on the plasma membrane of nucleated cancer cells [8]. In the cytosol, by supporting folding, refolding, and assembly of nascent polypeptides, HSP70s maintain protein homeostasis and thereby prevent protein aggregation, and it also assists the transport of other proteins across membranes [9]. Environmental stressors, such as nutrient [10] or ATP deprivation [11], thermal stress [12], ischemia [13], reactive oxygen species [10], and other free radicals [14], as well as physiological processes such as cell differentiation, maturation, and proliferation, induce the synthesis of the major stress-inducible Hsp70 in normal and tumor cells. Overexpression of Hsp70 in the cytosol of tumor cells promotes tumor growth and therapy resistance by activating anti-apoptotic and cytoprotective capacities [11,15]. Furthermore, it has been shown that multiple tumor types present Hsp70 on their plasma membrane [16], and an mHsp70 positive phenotype has been described for a large variety of highly aggressive tumor entities [17], including urological, lung, head, and neck tumors and tumors of the female reproductive tract such as ovarian, cervical carcinoma and breast cancers [15,18,19,20]. High levels of intracellular and mHsp70 are associated with resistance to standard therapies such as chemo- and radiotherapy and can enhance the invasive and metastatic potential [17,18,19]. It has also been shown that mHsp70 expression is a negative prognostic indicator for a number of other cancers [15].

Tumors expressing mHsp70 release extracellular lipid micro-vesicles (EVs), such as ectosomes and endosomes expressing Hsp70 on the surface [21]. The amount of vesicular Hsp70 in the blood, as measured using the compHsp70 ELISA [21], has been shown to be associated with the gross tumor volume (GTV) in patients with advanced non-small cell lung cancer (NSCLC) [22]. Although free Hsp70 mostly originates from inflamed and dying cells, vesicular Hsp70, which is found at high concentrations in the blood of tumor patients, is actively released by viable tumor cells expressing mHsp70 [21].

As the density of mHsp70 expression is higher on metastases compared to primary tumors [23], we hypothesized that mHsp70 also is expressed on the cell surface of circulating tumor cells (CTCs) [24], which are considered precursors of metastases. Although antibody-based approaches for isolating CTCs are typically based on antibodies recognizing Epithelial Cell Adhesion Molecule (EpCAM, CD326), the fact that EpCAM (CD326) is often downregulated after Epithelial-to-Mesenchymal Transition (EMT) [25,26], likely influences the effectiveness of such approaches. In contrast, we have demonstrated that the expression of mHsp70 remains unaffected after EMT [27]. As a consequence, we developed an antibody-based bead approach using a unique monoclonal antibody (mAb) that recognizes membrane-bound Hsp70 (cmHsp70.1) and demonstrated that this isolated higher numbers of CTCs from the peripheral blood of patients with metastatic cancer than an equivalent EpCAM mAb-based bead approach [27].

In our quest to develop approaches for better monitoring the effectiveness of an individual therapy and predicting disease recurrence, herein, we measured circulating levels of vesicular Hsp70 in patients with early breast cancer during the course of different therapies (radiation therapy, chemotherapy, anti-hormone therapy) and in the follow-up period up to six months. We also isolated and enumerated CTCs from the blood of patients with metastatic and non-metastatic endometrial, prostate, lung, and head and neck carcinoma using cmHsp70.1 or EpCAM mAb-based selection approaches. For the CTC isolation, patients with advanced tumor stages were chosen because of the known rarity of CTCs; typically, less than 10 CTCs/mL peripheral blood are present in the blood of patients with metastatic disease [28].

## 2. Materials and Methods

### 2.1. Study Design, Patients, and Sample Collection

This study analyzed frozen plasma samples from 35 female patients with non-metastatic breast cancer (T1/T2) from a case-control-study who were treated at the Klinikum rechts der Isar, Technische Universität München (TUM). Patients with secondary carcinoma, distant metastasis, previous radiation therapy, neoadjuvant chemotherapy, or prior breast cancer were excluded. All patients received breast-preserving surgery and, with the exception of two individuals, were treated with subsequent adjuvant radiation therapy. Depending on their hormone receptor status and tumor stage, patients received an adjuvant chemotherapy (FEC; 5-Fluoruracil, Epirubicin, Cyclophosphamide) and/or an anti-hormone therapy (Anastrozol, Arimidex, or Tamoxifen).

Patients with advanced metastatic castration-resistant prostate cancer (mCRPC) were prospectively enrolled in the biomarker trial “HSP70CTC” (NCT04628806). All patients were scheduled to receive [^177^Lu]-PSMA radioligand therapy at the Charité University Hospital, Berlin, Germany. EDTA blood samples (2 × 7.5 mL) were collected from these patients at diagnosis (*n* = 16). Samples were also collected from a cohort of patients with endometrial carcinoma (*n* = 3), lung cancer (*n* = 19), squamous cell carcinoma of the head and neck (SCCHN; *n* = 24), and from healthy individuals (*n* = 109). Approval for the blood sampling from patients and healthy individuals was obtained by the local Institutional Review Boards (Ethics Committee) of the Charité University Hospital, Berlin, and the Medical Faculty of Klinikum rechts der Isar, Technische Universität München, Germany, respectively. The study was conducted in accordance with the Declaration of Helsinki of 1975, and all participants signed an informed consent prior start of the study.

### 2.2. Measurement of Circulating Hsp70 Levels Using the compHSP70 ELISA [21]

Plasma was prepared from EDTA blood (S-Monovette, Sarstedt, Nümbrecht, Germany) by centrifugation at 1500× *g* for 15 min at room temperature and aliquots (300 μL) stored at −80 °C. For the compHsp70 ELISA, 96-well MaxiSorp Nunc-Immuno plates (Thermo, Rochester, NY, USA) were incubated with the cmHsp70.2 coating mAb (1 µg/mL; multimmune GmbH (Munich, Germany)) in sodium carbonate buffer (0.1 M sodium carbonate, 0.1 M sodium hydrogen carbonate, pH 9.6; Sigma-Aldrich (Darmstadt, Germany)) overnight at room temperature. Plates were washed with phosphate-buffered saline (PBS) (Life Technologies (Darmstadt, Germany)) containing 0.05% *v/v* Tween-20 (Calbiochem, Merck, Darmstadt, Germany) and blocked with liquid plate sealer (Candor Bioscience GmbH, Wangen i. Allgäu, Germany) for 30 min to prevent non-specific protein binding. After a washing step, plasma samples (100 µL) diluted 1:5 in StabilZyme Select Stabilizer (Diarect GmbH, Freiburg i. Breisgau, Germany) were added, as was a pre-diluted Hsp70 protein standard (0–100 ng/mL) and plates were incubated for 30 min at room temperature. After another washing step, plates were incubated for 30 min with biotinylated cmHsp70.1 detection mAb (multimmune GmbH, Munich, Germany; 200 ng/mL) dissolved in HRP-Protector (Candor Bioscience GmbH, Wangen i. Allgäu, Germany) and after a final washing step for another 30 min with 57 ng/mL Streptavidin (Senova GmbH, Weimar, Germany) in HRP-Protector^TM^ (Candor Bioscience GmbH, Wangern i. Allgäu, Germany). Colorimetric analysis was performed after incubation with the substrate reagent (100 µL) (BioFX TMB Super Sensitive One Component HRP Microwell Substrate, Surmodics, Inc., Eden Prairie, MN, USA) for 15 min. After stopping the colorimetric analysis by adding 2N H_2_SO_4_ (50 µL) the absorbance was read at 450 nm in a Microplate Reader (VICTOR X4 Multilabel Plate Reader, PerkinElmer, Waltham, MA, USA) and corrected by the absorbance at 570 nm. A market-ready Hsp70-exo ELISA kit that uses identical reagents is being manufactured by DRG Instruments GmbH, Marburg, Germany.

### 2.3. Isolation of CTCs with cmHsp70.1 and EpCAM Antibody-Coupled S-PluriSelect Beads

Isolation of CTCs was undertaken essentially as described previously [27]. Briefly, EDTA blood was incubated for 30 min with S-PluriSelect beads (PluriSelect Life Sciences, Leipzig, Germany) covalently coupled to the cmHsp70.1 (multimmune GmbH, Munich, Germany) or EpCAM (CD326, clone HEA125; Origene/Acris GmbH, Herford, Germany) mAbs under gentle rotation at room temperature. CTCs bound to the antibody-coupled beads were washed on a sterile filter with at least 20 mL wash buffer (PluriSelect Life Sciences). CTCs were detached from the beads by incubation with detachment buffer (PluriSelect Life Sciences) for 10 min and were then filtered through a sterile filter, washed in medium, and incubated overnight at 37 °C in a 48-well plate. After 24 h, CTCs were counted using a Zeiss Axiovert microscope (40× magnification) and kept in cell culture for additional counting after 5 to 7 days in cell culture.

### 2.4. Statistical Tests

The Kolmogorov–Smirnov test and the Shapiro–Wilk test showed that the Hsp70 concentrations in healthy donors and cancer patients were not normally distributed (*p* < 0.001 in each case). As a consequence, the Mann–Whitney U-test was used to compare two unrelated groups, and a one-factor analysis of variance (ANOVA) was used for comparing more than two unrelated groups. The Spearman correlation was used for calculating correlations between two metric variables.

## 3. Results

### 3.1. Hsp70 Concentrations in the Blood of Patients with Breast Cancer during Different Therapies and in the Follow-Up Period

Between 2013 and 2015, patients with localized unilateral breast cancer (*n* = 40) were treated in a curative intention trial. Eight years after the initial diagnosis of breast cancer, patients were contacted by a physician, and of the initial 40 patients, 35 patients agreed to provide information regarding their clinical course. Of these patients, 30 were in tumor stage T1 (a–c), and 5 in stage T2 at initial diagnosis. None of the patients had distant metastases, and all tumors were estrogen and progesterone receptor-positive (Table 1). An accepted limitation of the clinical trial was that no blood sample from before the start of therapy (surgical resection) was available from this patient cohort.

As summarized in Table 1, all patients received a breast-preserving surgery and, except for two individuals, subsequent radiotherapy with varying total doses ranging from 40 to 66 Gy according to conventional normo-fractionated or moderately hypofractionated schedules. Three patients were irradiated with 40 Gy, 27 patients with 60 Gy, and 3 patients with 66 Gy. Depending on tumor-specific characteristics, additional chemotherapy (FEC; 5-Fluoruracil, Epirubicin, Cyclophosphamide) and/or anti-hormone therapy (Anastrozol, Arimidex, or Tamoxifen) was given.

The first blood sample was taken postoperatively before the start of radiotherapy (**1**), then after 30 Gy (**2**), after the end of radiotherapy (**3**), six weeks after radiotherapy (**4**) and six months after radiotherapy (**5**) (Figure 1).

To investigate the potential effects of a specific therapy (radiation, chemotherapy, anti-hormone therapy) on circulating Hsp70 levels, Hsp70 was measured at the 5 indicated time-points in the recurrence-free patient cohort (*n* = 33) treated with or without a specific therapy. A comparison of the circulating Hsp70 levels in recurrence-free patients receiving irradiation (40, 60, 66 Gy; *n* = 31) or no radiation (*n* = 2) revealed no significant differences at any time-point (**1** *p* = 0.910, **2** *p* = 0.907, **3** *p* = 0.763, **4** *p* = 1.000, **5** *p* = 1.000; Mann–Whitney U-test). However, as shown in Figure 2A, the Hsp70 values of patients receiving radiotherapy always remained below those without radiotherapy, although the differences failed to reach statistical significance.

A comparison of the circulating Hsp70 levels in recurrence-free patients (*n* = 33) receiving additional chemotherapy (*n* = 27) or no chemotherapy (*n* = 6) showed similar results (Figure 2B). The mean Hsp70 concentrations in patients without chemotherapy remained unaltered in the range between 783.2 to 802.7 ng/mL throughout the whole study period, whereas the Hsp70 concentrations in the patients who were treated with chemotherapy remained constantly below these values ranging between 301.4 and 513.3 ng/mL. However, the differences of the values at all time-points were not of statistical significance (**1** *p* = 0.134, **2** *p* = 0.100, **3** *p* = 0.123, **4** *p* = 0.409, **5** *p* = 0.828, Mann–Whitney-U-test).

Although all patients showed a positive hormone receptor status (Estrogen receptor^+^, Progesterone receptor^+^), anti-hormone therapy was only given to 10 of the 33 patients. The Hsp70 concentrations in the recurrence-free patients with and without anti-hormone therapy were nearly identical until the end of radiotherapy (**1**–**3**) but dropped slightly during the anti-hormone therapy (**4**–**5**) (Figure 2C). However, due to the relatively low number of patients included in the study, statistical significance was not reached at any time-point (**1** *p* = 0.512, **2** *p* = 0.450, **3** *p* = 0.488, **4** *p* = 0.730, **5** *p* = 0.860, Mann–Whitney U-test).

### 3.2. Hsp70 Concentrations in the Blood of Patients with Breast Cancer with and without Recurrence

Of the 35 patients with stage T1 and T2 breast cancer, 33 patients remained recurrence-free, and there was no newly diagnosed disease of other origin eight years after diagnosis. However, one patient developed an endometrial carcinoma, and another patient a contralateral breast cancer which was diagnosed two years after the first diagnosis. To study the predictive value of circulating Hsp70 levels as an early tumor biomarker, Hsp70 concentrations of the 33 recurrence-free patients at the 5 different time-points were compared to those of the patient with the endometrial carcinoma (Figure 3A) and the patient with the contralateral breast cancer recurrence (Figure 3B). As shown in Figure 3A, the Hsp70 concentrations in the blood of the patient who developed an endometrial carcinoma two years after diagnosis of the breast cancer were found to be already elevated after the breast-conserving surgery (**1**) and increased further during radiotherapy (**2**). The values remained above the 95% confidence interval of the cohort of recurrence-free patients during radiotherapy (**2**) as well as in the follow-up period – a time-point when no endometrial carcinoma was diagnosed. The patient was irradiated with a total dose of 66 Gy but received no chemo- or anti-hormone therapy. The finding that a newly diagnosed, additional patient with endometrial carcinoma whose tumor has spread into the lymph nodes had very high Hsp70 levels in the circulation (1171 ng/mL) suggests that the elevated circulating Hsp70 levels shown in Figure 3A might originate from the endometrial carcinoma cells.

A comparison of the Hsp70 concentrations in the blood of another breast cancer patient who was diagnosed with a contralateral breast cancer two years after the first diagnosis by classical imaging methods with those who remained recurrence-free (*n* = 33) revealed a progressive increase in the circulating Hsp70 values which was already apparent during radiotherapy. It is known that highly aggressive tumors not only present Hsp70 on the plasma membrane at a high cell surface density [23] but also actively release Hsp70 in extracellular micro-vesicles (EVs), levels of which can be measured in the circulation using the compHsp70 ELISA [21]. As shown in Figure 3B, the Hsp70 concentrations in the patient with recurrent breast cancer increased from 512.4 ng/mL after surgical removal of the breast cancer (**1**) to 759.3 ng/mL after 30 Gy irradiation (**2**), to 1092.7 ng/mL after the end of radiotherapy (**3**), to 1816.7 ng/mL six weeks (**4**) and to 3101.7 ng/mL six months after the end of radiotherapy (**5**). Compared to all other recurrence-free patients, this patient was the only one whose Hsp70 values progressively increased. The Hsp70 concentrations of this patient were above the 95% confidence interval (CI) of the recurrence-free patient cohort after the end of radiotherapy (**3**–**5**). With respect to these findings, we speculate that the drastically elevated Hsp70 values in the circulation of the patient with recurrent breast cancer reflect viable tumor mass (also known as minimal residual disease, MRD) at an early stage before the tumor reached a size that could be diagnosed by classical imaging methods. The patient with recurrent disease was irradiated with a total dose of 60 Gy and was receiving chemotherapy.

### 3.3. Hsp70 as a Target for CTC Isolation in the Peripheral Blood of Patients with Metastatic Tumors

Next, we asked whether mHsp70 on tumor cells, which is a source of extracellular Hsp70 in circulation, could be used as a target for the isolation of circulating tumor cells (CTCs). Since the density of mHsp70 is higher on metastases than on primary tumors [23], and CTCs are considered precursors of metastases [24], we proposed that Hsp70 is present at high densities on the cell surface of CTCs. Most antibody-based CTC isolation systems are based on the expression of EpCAM (CD326) by CTCs [29]; however, EpCAM is often down-regulated after Epithelial-to-Mesenchymal Transition (EMT). In contrast, we have previously shown that mHsp70 expression remains stably high on the surface of CTCs after an artificially induced EMT with TGFβ [27]. With respect to these findings, we isolated CTCs from the blood of patients with mCRPC, metastatic and non-metastatic endometrial, lung cancer, and HNSCC using comparative Hsp70 and EpCAM mAb-based bead approaches, as described previously [27]. The patient characteristics and circulating Hsp70 values in these patients are summarized in Table 2.

As shown in Figure 4 and Table 2, circulating Hsp70 levels in patients with mCRPC (mean 272.8 ± 433.7 ng/mL; **** *p* < 0.0001; *n* = 16), endometrial carcinoma (mean 524.0 ± 578.3 ng/mL; ** *p* < 0.01; *n* = 3), lung cancer (mean 161.7 ± 253.9 ng/mL; *n* = 19) and head and neck carcinoma (mean 196.4±300.8 ng/mL; *** *p* < 0.001; *n* = 24) were significantly higher than those in a healthy control cohort (mean 35.5 ± 41.7 ng/mL; *n* = 109) (Mann–Whitney test). Hsp70 levels of a patient with metastatic endometrial carcinoma (1171 ng/mL; *n* = 1) were drastically higher than that of patients with non-metastasized endometrial carcinoma (200.4 ng/mL; *n* = 2).

A comparison of the number of CTCs that were isolated from the blood of patients with metastatic prostate cancer using the cmHsp70.1 (range 2–4000; mean 803 ± 1192) and EpCAM (range 0–2560; mean 393 ± 660) mAb-based bead approaches revealed significantly higher CTC counts (* *p* < 0.05) when using the cmHsp70.1 mAb-based approach (Figure 5A). The values derived with the EpCAM-based bead system are comparable to those CTC counts which have been isolated from patients with prostate cancer (mean 124 ± 400) using the FDA-approved EpCAM mAb-based CELLSEARCH^®^ system [30,31,32,33]. In a patient with metastatic endometrial carcinoma, the number of CTCs isolated with Hsp70 mAb-based and EpCAM mAb-based beads was 12,335 and 379, respectively (Figure 5B). When metastasized and non-metastasized endometrial carcinoma patients (*n* = 3) were compared, the number of CTCs was 4764 (range 467–12,935; mean 4764 ± 7079) vs. 399 (range 379–435; mean 399 ± 31), respectively (Figure 5B). Similar to the findings from patients with prostate carcinoma, higher CTC counts were obtained from patients with lung and head and neck cancer when using the cmHsp70.1 mAb-based bead approach. In patients with lung cancer (*n* = 12) the number of CTCs isolated with cmHsp70.1 mAb-coated beads was 924 (range 23–2456; mean 924 ± 876) vs. 820 (range 52–3611; mean 820 ± 1008), respectively (Figure 5C), and in patients with head and neck cancer (*n* = 16) the number of CTCs isolated using the cmHsp70.1 mAb-coated beads was 753 (range 0–2907; mean 753 ± 889) vs. 415 (range 0–1345; mean 415 ± 457.0), respectively (Figure 5D).

Figure 5E shows representative views of a DAPI (blue), cytokeratin (green), and CD45 (red) staining of CTCs derived from the blood of patients with metastatic prostate cancer following isolation using the cmHsp70.1 mAb-based bead approach. In both patients, CTCs were positively stained for cytokeratin (green). In one sample, the cytokeratin-positive CTC was localized in close proximity to a CD45 positively stained leukocyte (red). The nucleus of the cells was stained blue with DAPI.

## 4. Discussion

This study asked how radiation-, chemo-, or anti-hormone therapies affect levels of exosomal/microvesicular Hsp70 in the blood of recurrence-free patients with breast cancer in stage T1 and T2 during the therapy and in the follow-up period and whether increasing levels of exosomal Hsp70 could serve as a potential biomarker to monitor therapeutic response and/or predict subsequent therapy failure at an early time-point. Of the 35 patients studied, 33 patients remained recurrence-free 8 years after the first diagnosis of breast cancer, whereas one patient developed an endometrial carcinoma, and another patient developed a contralateral breast cancer.

The correlation between high expression of Hsp70 and clinical parameters such as diagnosis, prognosis, and response to therapy in different cancers has been shown for both intracellular [34,35,36,37,38,39] and extracellular Hsp70 levels [22,40,41,42]. As Hsp70 is overexpressed in a variety of tumors, it does not appear useful in diagnostic immunopathology for a specific tumor entity, as more specific and targeted markers to identify the lineage of cancer tissue exist [15]. Nevertheless, vesicular Hsp70 [43,44], which most likely originates from viable tumor cells and can be measured in the plasma by the compHsp70 ELISA, has great potential as a universal tumor marker in different tumor entities to make a significant contribution to the assessment of treatment response and might be useful [45,46,47].

To determine whether the plasma concentration of Hsp70 could be used as a predictor for response to a specific therapy, circulating Hsp70 values were compared in the blood of breast cancer patients with and without receiving radio-, chemo- and anti-hormone therapy. Although not statistically significant, all patients who received adjuvant radio- or chemotherapy showed consistently lower Hsp70 concentrations over the whole study period up to six months after therapy than the respective disease groups without therapy. During (time-points **4**, **5**) but not before or after anti-hormone therapy circulating Hsp70 values were slightly reduced in breast cancer patients.

Although patients with recurrence-free disease presented relatively constant Hsp70 values, the Hsp70 concentration of a patient who developed an endometrial carcinoma in the further course of the disease already increased while receiving radiotherapy. Since a patient with metastatic endometrial carcinoma showed very high circulating Hsp70 levels (Table 2), we speculate that the elevated Hsp70 levels originate from endometrial carcinoma cells in the breast cancer patient.

The Hsp70 values of another patient, who was diagnosed with a contralateral breast cancer two years after the first diagnosis, already showed a continuous increase of the Hsp70 levels after radiotherapy and in the follow-up period up to six months. Since the recurrent-free patients showed no increase in their circulating Hsp70 levels during the whole observation period, we speculate that the increase in circulating Hsp70 levels in the patients with recurrence or endometrial carcinoma indicates therapeutic failure. Due to the continuous increase in Hsp70 levels, which was already apparent at an early time-point (during radiotherapy, **2**), the recurrence could have been detected much earlier than with the imaging-based methods that are currently used for follow-up.

Previously, it was shown that levels of exosomal Hsp70 in the circulation reflect the mHsp70 status of the tumor [21,35]. Furthermore, we have previously shown, in mouse models and patients, that metastases exhibit a higher mHsp70 density than primary tumors [23]. Although free Hsp70 originates from apoptotic tumor cells [22], a constant increase in the levels of exosomal Hsp70, therefore, appears to indicate an increase in viable tumor cells in the context of recurrence or metastases. It is well established that the time of diagnosis influences and correlates with the survival rate, not only in the context of the initial diagnosis but also in the context of identifying the development of metastases which account for a large proportion (~90%) of cancer-specific mortality [48]. After completing the primary treatment, patients should participate in a follow-up program for at least ten years. In addition to medical history and physical examination, this includes image-based diagnostics [1,2,3,4]. However, the insufficient sensitivity of conventional imaging diagnostics to detect metastases at an early stage reflects an urgent clinical need for new approaches to detect recurrence and metastases early.

The repeated determination of Hsp70 in the blood of patients with different tumors provides an additional tool that might enable earlier detection of tumor recurrence in patients without radiation exposure by radiological imaging methods. Blood sampling is a minimally invasive method that can be repeated at any time and is well tolerated by patients. Monitoring the dynamics of Hsp70 levels in the blood using the compHsp70 ELISA at diagnosis, during therapy, and in the context of follow-up examinations has the potential to make a considerable contribution to the assessment of therapeutic response and to the prediction and identification of relapse. As no larger valid studies have been conducted on individualized risk-adapted follow-up programs [3,4], larger-scale studies to test the suitability of circulating Hsp70 as a biomarker for therapy failure in breast carcinoma patients and to define cut-off values are of great importance and urgency, as are equivalent studies in other cancer settings. Since Hsp70 is overexpressed and released by many different tumor entities [17,21,22], including breast, endometrial, prostate, lung, and head and neck carcinomas, as shown in this study, circulating Hsp70 values can have prognostic values in a large variety of different tumor entities [38,39,40,45,46].

In addition to circulating Hsp70, the enumeration of CTCs plays an important role in the categorization of cancer, patient prognosis, and the prediction of metastatic spread [47,48]. However, the rarity of this cell type in the peripheral blood and the loss of EpCAM expression on CTCs after EMT limits the capacity of EpCAM mAb-based approaches to effectively isolate CTCs [47]. Herein, we demonstrated that a CTC isolation system that targets mHsp70 on tumor cells [27] is superior to an EpCAM mAb-based system since at least equal, but mostly higher numbers of CTCs could be isolated from the peripheral blood of patients with metastatic prostate cancer who received a [^177^Lu]-PSMA-radioligand therapy later on [49], endometrial carcinoma [50], lung and squamous cell carcinoma of the head and neck. Moreover, elevated circulating Hsp70 levels in the blood, which are most likely derived from Hsp70 that is actively released in extracellular lipid micro-vesicles from viable tumor cells [21], do not negatively affect the mHsp70-targeted CTC isolation process.

## 5. Conclusions

Measuring Hsp70 concentrations and CTC counts in the circulation of patients with different tumor entities using the compHsp70 ELISA and an mHsp70-targeting bead approach has the potential to make a significant contribution in the early assessment of therapy response and prognosis (enumeration of CTCs) of patients with different tumor entities. Further studies with higher patient numbers are warranted to confirm and validate the results.

## 6. Patents

The compHsp70 ELISA is patented by multimmune GmbH, Munich, Germany (US 11,460,472—other applications pending).

## Figures and Tables

**Figure 1 biomedicines-11-02276-f001:**
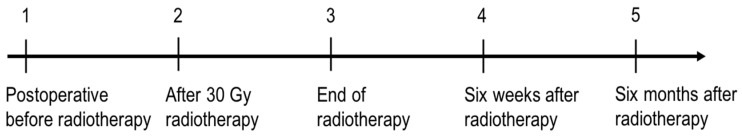
Time-points 1 to 5 of blood sampling of patients with breast cancer after surgery.

**Figure 2 biomedicines-11-02276-f002:**
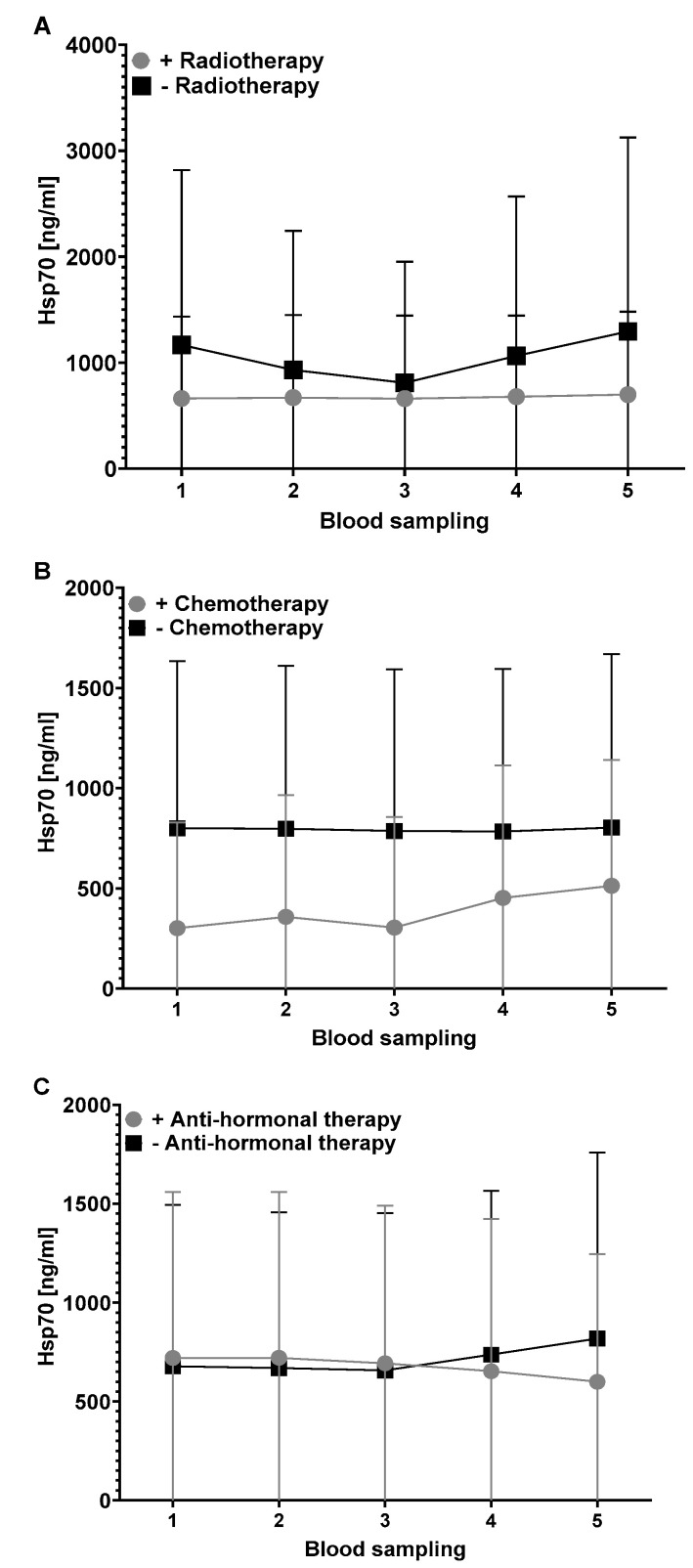
Hsp70 concentrations in the blood of recurrence-free patients with (*n* = 31) and without receiving radiotherapy (*n* = 2) (**A**), with (*n* = 6) and without (*n* = 27) receiving chemotherapy (**B**) and with (*n* = 12) and without (*n* = 21) receiving anti-hormone therapy (**C**).

**Figure 3 biomedicines-11-02276-f003:**
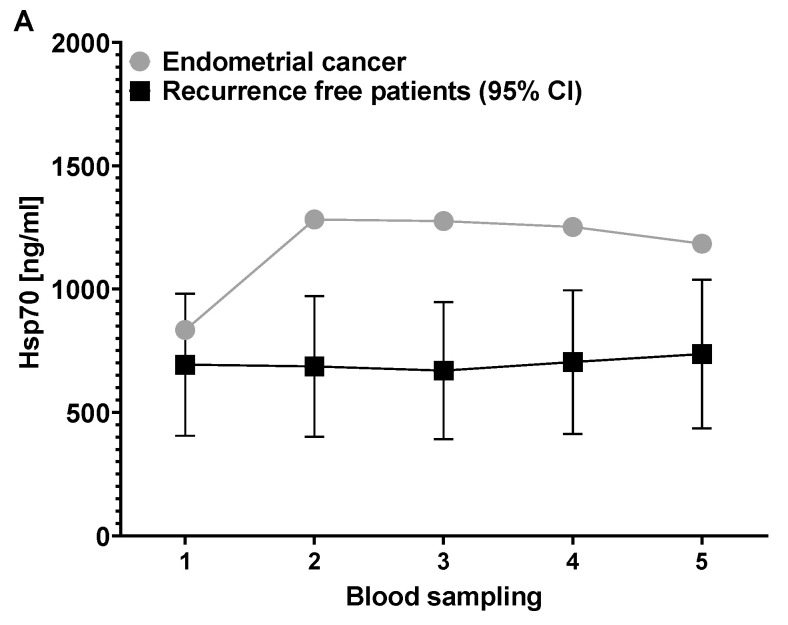
Hsp70 concentrations in the blood of a patient with endometrial cancer (*n* = 1) (**A**) and a patient with contralateral recurrent breast cancer (*n* = 1) (**B**) compared to recurrence-free (*n* = 33) patients. Abbreviation: CI, confidence interval.

**Figure 4 biomedicines-11-02276-f004:**
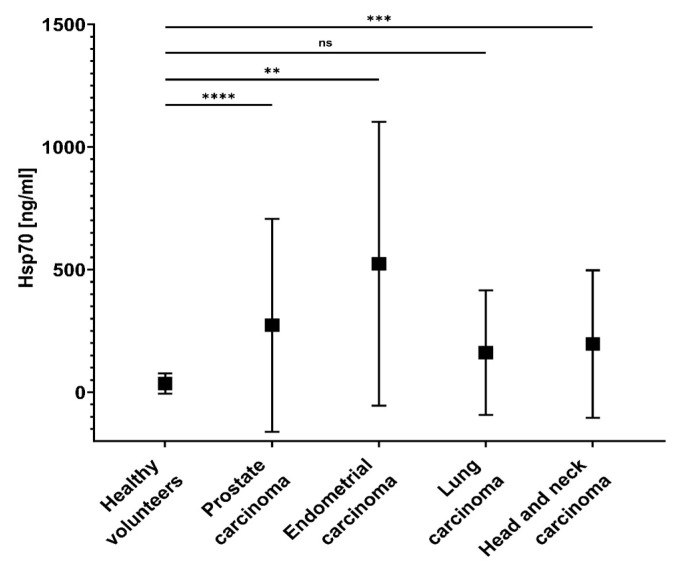
Circulating Hsp70 levels in the peripheral blood of healthy volunteers (*n* = 109) and patients with metastatic prostate carcinoma (mCRPC; *n* = 16), metastatic (*n* = 1), and non-metastatic (*n* = 2) endometrial carcinoma (*n* = 1), lung carcinoma (*n* = 19), head and neck carcinoma *n* = 24) before the start of any therapy as determined with the compHsp70 ELISA. Mann–Whitney test; statistical differences **** *p* < 0.0001; *** *p* < 0.001; ** *p* < 0.01; ns not significant.

**Figure 5 biomedicines-11-02276-f005:**
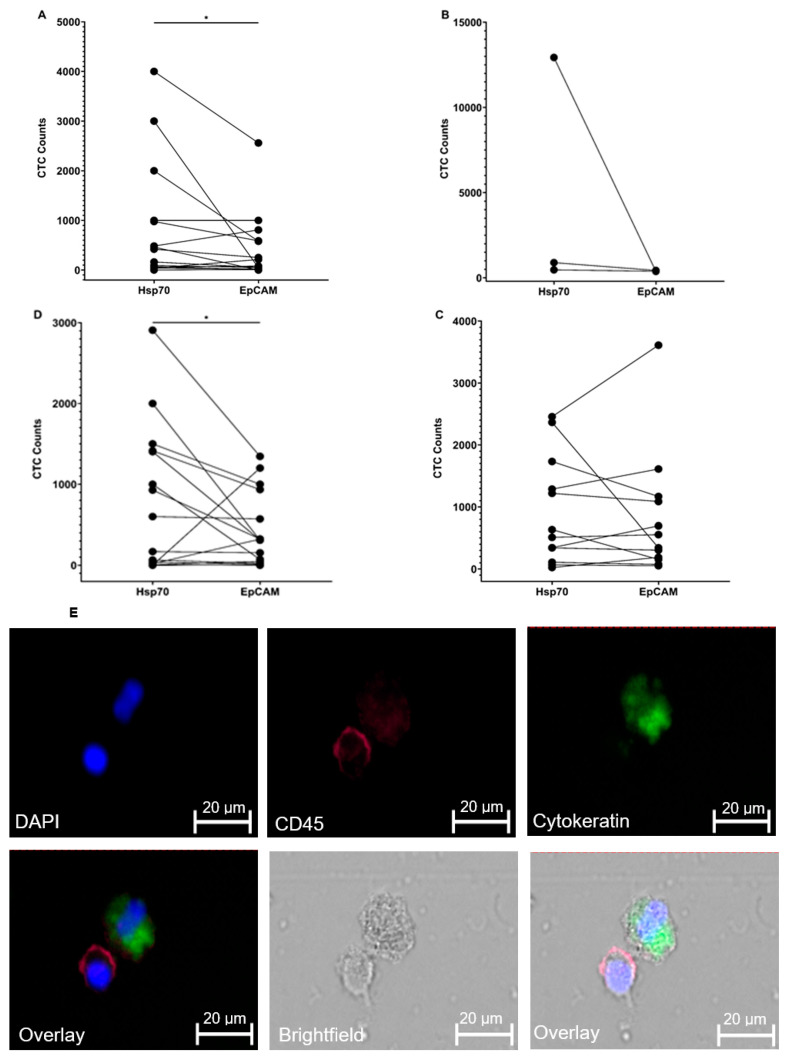
Enumeration of CTCs in the peripheral blood of patients with metastatic prostate cancer (*n* = 16) (**A**), a patient with metastatic endometrial carcinoma (*n* = 1) and 2 patients with non-metastatic endometrial carcinoma (**B**), lung carcinoma (*n* = 12) and head and neck carcinoma (*n* = 16) as isolated using cmHsp70.1 and EpCAM mAb-conjugated beads (**C**,**D**). Representative photomicrographs of CTCs and a leukocyte derived from patients with metastatic prostate carcinoma following CTC isolation using cmHsp70.1 mAb-conjugated beads (**E**). DAPI (blue) and fluorescence-labeled antibodies directed against CD45 (red) and cytokeratin (green) identify the nucleus, leukocytes, and epithelial cells. Scale bar, 20 µm. * *p* < 0.05.

**Table 1 biomedicines-11-02276-t001:** Patient characteristics, therapies, and clinical outcomes.

Breast Cancer Patients 8 Years after Diagnosis	(*n* = 35)	
Grade		
T1 (a–c)	30	
T2	5	
N0	30	
N1	5	
M0	35	
G1	7	
G2	26	
G3	2	
Therapy	Yes	No
Surgery	35	0
Radiotherapy	33	2
40 Gy	3	
60 Gy	24	
66 Gy	6	
Chemotherapy	7	28
Anti-hormone therapy	12	23
Estrogen receptor^+^	35	0
Progesterone receptor^+^	35	0
Clinical status after 8 years	Recurrence-free	Contralateral recurrence/ Endometrial cancer
	33	2

**Table 2 biomedicines-11-02276-t002:** Characteristics of patients with metastatic (*n* = 1) and non-metastatic (*n* = 2) endometrial carcinoma, metastatic castration-resistant prostate carcinoma (mCRPC; *n* = 16) and tumor response 3 months after [^177^Lu]-PSMA-radioligand therapy, lung carcinoma (*n* = 19) and squamous cell carcinoma of the head and neck (HNSCC; *n* = 24) and circulating Hsp70 values.

Tumor	Number of Patients	Hsp70 Values (ng/mL)
**Metastatic endometrial** **carcinoma**	1	1171
**Non-metastatic endometrial carcinoma**	2	200.4
**Metastatic castration resistant prostate carcinoma (mCRPC)**	16	272.8 ± 433.7
N0	1	
N1	13	
Unknown	2	
M1a	2	
M1b	10	
M1c	4	
**Therapies**		
Pretreatments for metastatic prostate cancer		
Next generation hormonal treatment	16	
Taxane-based chemotherapy	9	
Best treatment response 3 months after [^177^Lu]-PSMA- radioligand therapy		
PR	4	
SD	3	
PD	7	
Lost to follow-up	2	
**Tumor**	**Number of patients**	**Hsp70 values** **(ng/mL)**
**Lung carcinoma**	19	161.7 ± 253.9
IA	4	
IB	2	
IIB	3	
IIIA	3	
IVA/B	4	
Pulmonary metastases		
renal/urothelial	2	127.5
Carcinoid	1	1.23
**Head and neck carcinoma**	24	196.4 ± 300.8
Oral cavity	6	
Oropharynx	8	
Hypopharynx	3	
Tonsil	3	
Thyroid	1	
Uvula	1	
Tongue	2	

Abbreviations: PR, partial response; SD, stable disease; PD, progressive disease.

## Data Availability

The data presented in this study are available upon request from the corresponding author.

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
