# Peer review of "Hsp70—A Universal Biomarker for Predicting Therapeutic Failure in Human Female Cancers and a Target for CTC Isolation in Advanced Cancers"

_biomedicines, 2023, doi:10.3390/biomedicines11082276_

Round 1
Reviewer 1 Report
The authors propose blood Hsp70 measurement for following tumor progression/recurrence. In addition, they demonstrate that tumor cell surface Hsp70-based selection might be used to enrich and enumerate CTC. The idea is original and interesting, the experimental part is correct, but the message is compromised by the small sample sizes. The idea of the authors would be worth some extra experimental work. The comparison of the CTC numbres obtained by EpCAM- and Hsp70-based selection is also intriguing; the 32-fold difference between the CTC numbers established by the two different methods would warrant some additional analysis.
Author Response
The authors want to thank the reviewer for the constructive comments. In the meantime we had the opportunity to study the blood of patients with different tumor entities. The data were included into the Ms as recommended. We have included the data as a new Figure 4 and new Figure 5.
The English was revised by a native Englsih speaking co-author.
Reviewer 2 Report
This manuscript “Hsp70 – a Universal Biomarker for Predicting Therapy Failure 2 in Human Female Cancers and a Target for CTC Isolation in Metastatic Prostate Cancer” by A. Xanthopoulos et al. describes Heat shock protein 70 (Hsp70)-based selection system appears superior to an EpCAM-37 based approach in the isolation of CTCs from patients with metastatic cancer. The manuscript seems good as Hsp70 can be used as a new biomarker. Add the possibility for other cancers.
OK
Author Response
The authors want to thank the reveiwer for constructive suggestions. Since Hsp70 is overexpressed and presented on the cell surface of a large variety of different tumor entities it is very likely that circulating Hsp70 and membrane bound Hsp70 can be used as a tumor biomarker for predicting response and for the isolation of CTCs in different tumor entities. Additional data on Hsp70 values and CTC counts have been included for other tumor entities. The data are shown in a new Figure 4 and Figure 5 in the revised version of the Ms. Since these data were produced by additional students the number of authors has been extended.
The English was revised by a native English speaking co-author.
Round 2
Reviewer 1 Report
Adding supplementary data and explanations, the authors considerably improved the paper that is now suitable for publication.